# Scientific Language Models for Biomedical Knowledge Base Completion: An Empirical Study

**Rahul Nadkarni**[1]                                    RAHULN@CS.WASHINGTON.EDU
**David Wadden**[1]                                    DWADDEN@CS.WASHINGTON.EDU
**Iz Beltagy**[2]                                            BELTAGY@ALLENAI.ORG
**Noah A. Smith**[1,2]                                    NASMITH@CS.WASHINGTON.EDU
**Hannaneh Hajishirzi**[1,2]                            HANNANEH@CS.WASHINGTON.EDU
**Tom Hope**[1,2]                                              TOMH@ALLENAI.ORG
[1]*Paul G. Allen School of Computer Science & Engineering, University of Washington*
[2]*Allen Institute for Artificial Intelligence (AI2)*

## Abstract

Biomedical knowledge graphs (KGs) hold rich information on entities such as diseases, drugs, and genes. Predicting missing links in these graphs can boost many important applications, such as drug design and repurposing. Recent work has shown that general-domain language models (LMs) can serve as "soft" KGs, and that they can be fine-tuned for the task of KG completion. In this work, we study *scientific* LMs for KG completion, exploring whether we can tap into their latent knowledge to enhance biomedical link prediction. We evaluate several domain-specific LMs, fine-tuning them on datasets centered on drugs and diseases that we represent as KGs and enrich with textual entity descriptions. We integrate the LM-based models with KG embedding models, using a router method that learns to assign each input example to either type of model and provides a substantial boost in performance. Finally, we demonstrate the advantage of LM models in the inductive setting with novel scientific entities. Our datasets and code are made publicly available.[1]

## 1. Introduction

Understanding complex diseases such as cancer, HIV, and COVID-19 requires rich biological, chemical, and medical knowledge. This knowledge plays a vital role in the process of discovering therapies for these diseases — for example, identifying targets for drugs [Lindsay, 2003] requires knowing what genes or proteins are involved in a disease, and designing drugs requires predicting whether a drug molecule will interact with specific target proteins. In addition, to alleviate the great costs of designing new drugs, drug repositioning [Luo et al., 2021] involves identification of *existing* drugs that can be re-purposed for other diseases. Due to the challenging combinatorial nature of these tasks, there is need for automation with machine learning techniques. Given the many links between biomedical entities, recent work [Bonner et al., 2021a,b] has highlighted the potential benefits of *knowledge graph* (KG) data representations, formulating the associated tasks as *KG completion* problems — predicting missing links between drugs and diseases, diseases and genes, and so forth.

The focus of KG completion work — in the general domain, as well as in biomedical applications — is on using graph structure to make predictions, such as with KG embedding

---

1. https://github.com/rahuln/lm-bio-kgc

(KGE) models and graph neural networks [Zitnik et al., 2018, Chang et al., 2020]. In parallel, recent work in the general domain has explored the use of pretrained language models (LMs) as "soft" knowledge bases, holding factual knowledge latently encoded in their parameters [Petroni et al., 2019, 2020]. An emerging direction for using this information for the task of KG completion involves fine-tuning LMs to predict relations between pairs of entities based on their textual descriptions [Yao et al., 2019, Kim et al., 2020, Wang et al., 2021, Daza et al., 2021]. In the scientific domain, this raises the prospect of using LMs trained on millions of research papers to tap into the scientific knowledge that may be embedded in their parameters. While this text-based approach has been evaluated on general domain benchmarks derived from WordNet [Miller, 1995] and Freebase [Bollacker et al., 2008], to our knowledge it has not been applied to the task of scientific KG completion.

**Our contributions.** We perform an extensive study of LM-based KG completion in the biomedical domain, focusing on three datasets centered on drugs and diseases, two of which have not been used to date for the KG completion task. To enable exploration of LM-based models, we collect missing entity descriptions, obtaining them for over 35k entities across all datasets. We evaluate a range of KGE models and *domain-specific* scientific LMs pretrained on different biomedical corpora [Beltagy et al., 2019, Lee et al., 2020, Alsentzer et al., 2019, Gu et al., 2020]. We conduct analyses of predictions made by both types of models and find them to have complementary strengths, echoing similar observations made in recent work in the general domain [Wang et al., 2021] and motivating integration of both text and graph modalities. Unlike previous work, we train a router that selects for each input instance which type of model is likely to do better, finding it to often outperform average-based ensembles. Integration of text and graph modalities provides substantial relative improvements of 13–36% in mean reciprocal rank (MRR), and routing across multiple LM-based models further boosts results. Finally, we demonstrate the utility of LM-based models when applied to entities unseen during training, an important scenario in the rapidly evolving scientific domain. Our hope is that this work will encourage further research into using scientific LMs for biomedical KG completion, tapping into knowledge embedded in these models and making relational inferences between complex scientific concepts.

## 2. Task and Methods

We begin by presenting the KG completion task and the approaches we employ for predicting missing links in biomedical KGs, including our model integration and inductive KG completion methodologies. An overview of our approaches is illustrated in Figure 1.

### 2.1 KG Completion Task

Formally, a KG consists of entities $\mathcal{E}$, relations $\mathcal{R}$, and triples $\mathcal{T}$ representing *facts*. Each triple $(h, r, t) \in \mathcal{T}$ consists of head and tail entities $h, t \in \mathcal{E}$ and a relation $r \in \mathcal{R}$. An entity can be one of many types, with the type of an entity $e$ denoted as $T(e)$. In our setting, each entity is also associated with some text, denoted as text$(e)$ for $e \in \mathcal{E}$. This text can be an entity name, description, or both; we use the entity's name concatenated with its description when available, or just the name otherwise. For example, the fact (*aspirin,*

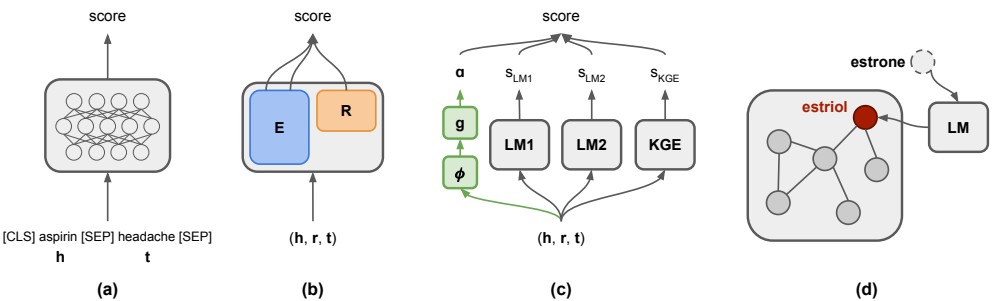

Figure 1: Illustration of the main methods we apply for biomedical KG completion: (a) LM fine-tuning; (b) KGE models; (c) an approach that combines both; and (d) using an LM to impute missing entities in a KGE model.

*treats, headache*) might be an $(h, r, t)$ triple found in a biomedical KG that relates drugs and diseases, with the head and tail entities having types $T(h) = drug$ and $T(t) = disease$.

The task of *KG completion* or *link prediction* involves receiving a triple $(h, r, ?)$ (where ? can replace either the head or tail entity) and scoring all candidate triples $\{(h, r, t') \mid t' \in \mathcal{S}\}$ such that the correct entity that replaces ? has the highest score. For the example listed above, a well-performing model that receives the incomplete triple (*aspirin, treats*, ?) should rank the tail entity *headache* higher than an incorrect one such as *diabetes*. $\mathcal{S}$ can be the entire set of entities (i.e., $\mathcal{S} = \mathcal{E}$) or some fixed subset. In the *transductive* setting, the set of facts $\mathcal{T}$ is split into a training set $\mathcal{T}_{\text{train}}$ and a test set $\mathcal{T}_{\text{test}}$ such that all positive triples in the test set contain entities seen during training. In contrast, for *inductive* KG completion the triples in the test set may contain entities not seen during training (see Section 2.4).

## 2.2 Methods

**Ranking-based KG completion.** Each KG completion model in our experiments learns a function $f$ that computes a ranking score $s = f(x)$ for a given triple $x = (h, r, t)$. Models are trained to assign a high ranking score to correct positive triples from the set of known facts $\mathcal{T}$ and a low ranking score to triples that are likely to be incorrect. To do so, we use the max-margin loss function $\mathcal{L}_{\text{rank}}(x) = \frac{1}{N} \sum_{i=1}^{N} \max(0, \lambda - f(x) + f(x_i'))$, where $\lambda$ is a margin hyperparameter, $x \in \mathcal{T}$ is a known positive triple in the KG, and $x_i'$ is a negative triple constructed by randomly corrupting either the head or tail entity of $x$ with an entity of the same type.

**KG embedding (KGE) models.** For each entity $e \in \mathcal{E}$ and each relation $r \in \mathcal{R}$, KG embedding (KGE) models learn a vector representation $E(e) \in \mathbb{R}^m$ and $R(r) \in \mathbb{R}^n$. For a given triple $(h, r, t)$, each model computes the ranking score $f(h, r, t)$ as a simple function of these embeddings (Figure 1b). We include a variety of different KGE models in our experiments, including TransE [Bordes et al., 2013], DistMult [Yang et al., 2015], ComplEx [Trouillon et al., 2016], and RotatE [Sun et al., 2019].

**LM-based models.** KGE methods do not capture the rich information available from textual descriptions of nodes. To address this limitation, previous KG completion ap-

proaches have incorporated textual representations [Toutanova et al., 2015, Wang and Li, 2016], most recently with approaches such as KG-BERT [Yao et al., 2019] that fine-tune the BERT language model (LM) [Devlin et al., 2019] for the task of KG completion. Our focus in this work is on LMs pretrained on corpora of biomedical documents (e.g., PubMedBERT [Gu et al., 2020]; see Appendix B.1.2 for full details). To score a triple using an LM, we use a cross-encoder approach [Yao et al., 2019, Kim et al., 2020] (Fig. 1a), where we encode the text of the head and tail entities together along with the appropriate special tokens. Specifically, a triple $(h, r, t)$ is encoded as $v = \text{LM}(\texttt{[CLS]} \ \text{text}(h) \ \texttt{[SEP]} \ \text{text}(t) \ \texttt{[SEP]})$, where $v$ is the contextualized representation of the \texttt{[CLS]} token at the last layer.[2] We then apply an additional linear layer with a single output dimension to $v$ to compute the ranking score for the triple ($f(x) = W_{\text{rank}} v \in \mathbb{R}$), and train the LM with the same max-margin loss. Recent work on applying BERT for KG completion on general domain benchmarks has shown that multi-task training improves performance [Wang et al., 2021, Kim et al., 2020]. We use the approach of Kim et al. [2020] and incorporate two additional losses for each LM: a binary triple classification loss to identify if a triple is positive or negative, and a multi-class relation classification loss.[3]

## 2.3 Integrating KGE and LM: Model Averaging vs. Routing

Previous work using text for KG completion on general domain benchmarks has demonstrated the benefit of combining KGE and text-based models [Xie et al., 2016, Wang et al., 2021]. We study integration of graph-based and text-based methods (Figure 1c), exploring whether learning to route input instances adaptively to a *single* model can improve performance over previous approaches that compute a weighted average of ranking scores [Wang et al., 2021]. We also explore the more general setup of combining more than two models.

More formally, for a given triple $x = (h, r, t)$, let $\phi(x)$ be its feature vector. We can learn a function $g(\phi(x))$ that outputs a set of weights $\boldsymbol{\alpha} = [\alpha_1, \ldots, \alpha_k], \sum_i \alpha_i = 1, \alpha_i > 0 \ \forall i$. These weights can be used to perform a weighted average of the ranking scores $\{s_1, \ldots, s_k\}$ for a set of $k$ models we wish to combine, such that the final ranking score is $s = \sum_i \alpha_i s_i$. We use a variety of graph-, triple-, and text-based features to construct the feature vector $\phi(x)$ such as node degree, entity and relation type, string edit distance between head and tail entity names, and overlap in graph neighbors of head and tail nodes. We explore these features further in Section 4.1, and provide a full list in Appendix B.1.3 (Table 6).

For the function $g(\cdot)$, we experiment with an **input-dependent weighted average** that outputs arbitrary weights $\boldsymbol{\alpha}$ and a **router** that outputs a constrained $\boldsymbol{\alpha}$ such that $\alpha_i = 1$ for some $i$ and $\alpha_j = 0, \forall j \neq i$ (i.e., $\boldsymbol{\alpha}$ is a one-hot vector).[4] In practice, we implement the router as a classifier which selects a single KG completion model for each example by training it to predict which model will perform better.[5] For the input-dependent weighted average we train a multilayer perceptron (MLP) using the max-margin ranking loss. We train all models on the validation set and evaluate on the test set for each dataset.

---

2. We experiment with encoding the relation text as well, but find that this did not improve performance.

3. See details in Appendix B. Wang et al. [2021] omit the relation classification loss and use a bi-encoder; we find that both of these modifications reduce performance in our setting.

4. We also try a global weighted average with a single set of weights; see Appendix B.1.3 for details.

5. We explore a range of methods for the router's classifier, with the best found to be gradient boosted decision trees (GBDT) and multilayer perceptrons (MLP).

| | | | #Positive Edges | | | |
|---|---|---|---|---|---|---|
| Dataset | #Entities | #Rel | Train | Dev. | Test | Avg. Desc. Length |
| RepoDB | 2,748 | 1 | 5,342 | 667 | 668 | 49.54 |
| Hetionet (our subset) | 12,733 | 4 | 124,544 | 15,567 | 15,568 | 44.65 |
| MSI | 29,959 | 6 | 387,724 | 48,465 | 48,465 | 45.13 |
| WN18RR | 40,943 | 11 | 86,835 | 3,034 | 3,134 | 14.26 |
| FB15k-237 | 14,541 | 237 | 272,115 | 17,535 | 20,466 | 139.32 |

Table 1: Statistics for our datasets and a sample of general domain benchmarks.

When performing ranking evaluation, we use the features $\phi(x)$ of each positive example to compute the weights $\boldsymbol{\alpha}$, then apply the same weights to all negative examples ranked against that positive example.

## 2.4 Inductive KG Completion

KGE models are limited to the *transductive* setting where all entities seen during evaluation have appeared during training. *Inductive* KG completion is important in the biomedical domain, where we may want to make predictions on novel entities such as emerging biomedical concepts or drugs/proteins mentioned in the literature that are missing from existing KGs. Due to their ability to form compositional representations from entity text, LMs are well-suited to this setting. In addition to using LMs fine-tuned for KGC, we try a simple technique using LMs to "fill in" missing KGE embeddings without explicitly using the LM for prediction (Fig. 1d). Given a set of entities $\mathcal{E}$ for which a KGE model has trained embeddings and a set of unknown entities $\mathcal{U}$, for each $e \in \mathcal{E} \cup \mathcal{U}$ we encode its text using an LM to form $v_e = \text{LM}(\texttt{[CLS]} \text{ text}(e) \texttt{ [SEP]}), \forall e \in \mathcal{E} \cup \mathcal{U}$, where $v_e$ is the $\texttt{[CLS]}$ token representation at the last layer. We use the cosine similarity between embeddings to replace each unseen entity's embedding with the closest trained embedding as $E(u) = E(\underset{e \in \mathcal{E}}{\text{argmax}} \text{ cos-sim}(v_e, v_u))$ where $e$ is of the same type as $u$, i.e., $T(e) = T(u)$.

## 3. Experimental Setup

### 3.1 Datasets

We use three datasets in the biomedical domain that cover a range of sizes comparable to existing general domain benchmarks, each pooled from a broad range of biomedical sources. Our datasets include **RepoDB** [Brown and Patel, 2017], a collection of drug-disease pairs intended for drug repositioning research; **MSI** (multiscale interactome; [Ruiz et al., 2021]), a recent network of diseases, proteins, genes, drug targets, and biological functions; and **Hetionet** [Himmelstein and Baranzini, 2015], a heterogeneous biomedical knowledge graph which following Alshahrani et al. [2021] we restrict to interactions involving drugs, diseases, symptoms, genes, and side effects.[6] Statistics for all datasets and a sample of popular general domain benchmark KGs can be found in Table 1.

---

6. More information on each dataset is available in Appendix A.1.

While Hetionet has previously been explored for the task of KG completion as link prediction using KGE models (though not LMs) [Alshahrani et al., 2021, Bonner et al., 2021b], to our knowledge neither RepoDB nor MSI have been represented as KGs and used for evaluating KG completion models despite the potential benefits of this representation [Bonner et al., 2021a], especially in conjunction with textual information. In order to apply LMs to each dataset, we scrape entity names (when not provided by the original dataset) as well as descriptions from the original online sources used to construct each KG (see Table 5 in the appendix).

We construct an 80%/10%/10% training/development/test transductive split for each KG by removing edges from the complete graph while ensuring that all nodes remain in the training graph. We also construct inductive splits, where each positive triple in the test test has one or both entities unseen during training.

### 3.2 Pretrained LMs and KGE Integration

We experiment with several LMs pretrained on biomedical corpora (see Table 2 and Appendix B.1.2). For each LM that has been fine-tuned for KG completion, we add the prefix "KG-" (e.g., KG-PubMedBERT) to differentiate it from the base LM. We use the umbrella term "model integration" for both model averaging and routing, unless stated otherwise.

**Model integration.** We explore integration of all pairs of KGE models as well as each KGE model paired with KG-PubMedBERT. This allows us to compare the effect of integrating pairs of KG completion models in general with integrating graph- and text-based approaches. For all pairs of models, we use the router-based and input-dependent weighted average methods. We also explore combinations of multiple KGE models and LMs, where we start with the best pair of KG-PubMedBERT and a KGE model based on the validation set and add either KG-BioBERT or the best-performing KGE model (or the second-best, if the best KGE model is in the best pair with KG-PubMedBERT).

### 3.3 Evaluation

At test time, each positive triple is ranked against a set of negatives constructed by replacing either the head or tail entity by a fixed set of entities of the same type. When constructing the edge split for each of the three datasets, we generate a fixed set of negatives for every positive triple in the validation and test sets, each corresponding to replacing the head or tail entity with an entity of the same type and filtering out negatives that appear as positive triples in either the training, validation, or test set (exact details in Appendix A.4). For each positive triple, we use its rank to compute the mean reciprocal rank (MRR), Hits@3 (H@3), and Hits@10 (H@10) metrics.

## 4. Experimental Results

### 4.1 Transductive Link Prediction Results

We report performance on the link prediction task across all datasets and models in Table 2. While LMs perform competitively with KGE models and even outperform some, they generally do not match the best KGE model on RepoDB and MSI. This echoes re-

| | | RepoDB | | | Hetionet | | | MSI | | |
|---|---|---|---|---|---|---|---|---|---|---|
| | | MRR | H@3 | H@10 | MRR | H@3 | H@10 | MRR | H@3 | H@10 |
| KGE | ComplEx | 62.3 | 71.1 | 85.6 | 45.9 | 53.6 | 77.8 | 40.3 | 44.3 | 57.5 |
| | DistMult | 62.0 | 70.4 | 85.2 | 46.0 | 53.5 | 77.8 | 29.6 | 34.1 | 53.6 |
| | RotatE | 58.8 | 65.9 | 79.8 | 50.6 | 58.2 | 79.3 | 32.4 | 35.3 | 49.8 |
| | TransE | 60.0 | 68.6 | 81.1 | 50.2 | 58.0 | 79.8 | 32.7 | 36.5 | 53.8 |
| LM (fine-tuned) | RoBERTa | 51.7 | 60.3 | 82.3 | 46.4 | 53.6 | 76.9 | 30.1 | 33.3 | 50.6 |
| | SciBERT | 59.7 | 67.6 | 88.5 | 50.3 | 57.1 | 79.1 | 34.2 | 37.9 | 55.0 |
| | BioBERT | 58.2 | 65.8 | 86.8 | 50.3 | 57.5 | 79.4 | 33.4 | 37.1 | 54.8 |
| | Bio+ClinicalBERT | 55.7 | 64.0 | 84.1 | 43.6 | 49.1 | 72.6 | 32.6 | 36.1 | 53.5 |
| | PubMedBERT-abs | 60.8 | 70.7 | 89.5 | 50.8 | 58.0 | 80.0 | 34.3 | 38.0 | 55.3 |
| | PubMedBERT-full | 59.9 | 69.3 | 88.8 | 51.7 | 58.7 | 80.8 | 34.2 | 37.7 | 55.1 |
| Two models (router) | Best pair of KGE | 62.2 | 70.4 | 83.7 | 56.1 | 65.5 | 85.4 | 45.2 | 50.6 | 66.2 |
| | Best KGE + LM | 70.6 | 80.3 | 94.3 | 59.7 | 68.6 | 87.2 | 48.5 | 54.4 | 70.1 |
| Two models (input-dep. avg.) | Best pair of KGE | 65.2 | 74.3 | 87.6 | 65.3 | 75.3 | 90.2 | 39.8 | 44.9 | 62.0 |
| | Best KGE + LM | 65.9 | 74.4 | 91.5 | **70.3** | **78.7** | **92.2** | 40.6 | 44.6 | 61.2 |
| Three models (router) | 2 KGE + 1 LM | **72.7** | 81.6 | 95.2 | 62.6 | 71.7 | 89.4 | 50.9 | **57.1** | **73.2** |
| | 1 KGE + 2 LM | 72.1 | **82.5** | **95.7** | 62.1 | 71.9 | 89.5 | **51.2** | 57.0 | 73.0 |

Table 2: KG completion results. All values are in the range [0, 100], higher is better. Underlined values denote the best result within a model category (KGE, LM, two models with router, two models with input-dependent weighted average, three models with router), while bold values denote the best result for each dataset.

sults in the general domain for link prediction on subsets of WordNet and Freebase [Yao et al., 2019, Wang et al., 2021]. On all datasets and metrics, the best-performing LM is KG-PubMedBERT, which aligns with results for natural language understanding tasks over biomedical text [Gu et al., 2020]. The biomedical LMs also generally outperform KG-RoBERTa, illustrating the benefit of in-domain pretraining even in the KG completion setting.

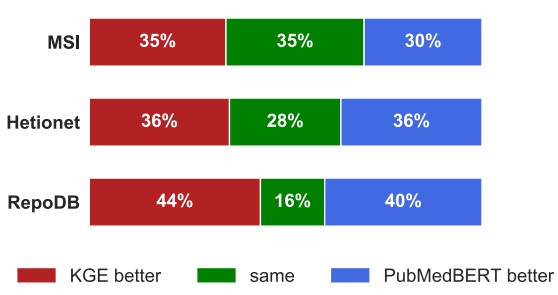

Figure 2: Fraction of test set examples where each model performs better.

**Comparing model errors.** By examining a selected set of examples in Table 3, we can observe cases where information in text provides LMs an advantage and where a lack of context favors KGE models. KG-PubMedBERT is able to make connections between biomedical concepts – like the fact that a disease that affects the *stomach* might cause *weight loss* – and align related concepts expressed with different terminology – like connecting *antineoplastic* with *cancer* (a type of *neoplasm*), or recognizing that an *echocardiogram* is a technique for imaging the *heart*. In contrast, RotatE offers an advantage when the descriptions do not immediately connect the two terms (*mediastinal cancer*, *hoarseness*), where a description

| Relation | RotatE better | KG-PubMedBERT better |
|---|---|---|
| Disease *presents* Symptom | Disease: **mediastinal cancer**; a cancer in the mediastinum.
Symptom: **hoarseness**; a deep or rough quality of voice. | Disease: **stomach cancer**; a gastrointestinal cancer in the stomach.
Symptom: **weight loss**; decrease in existing body weight. |
| Compound *treats* Disease | Compound: **methylprednisolone**; a prednisolone derivative glucocorticoid with higher potency.
Disease: **allergic rhinitis**; a rhinitis that is an allergic inflammation and irritation of the nasal airways. | Compound: **altretamine**; an alkylating agent proposed as an antineoplastic.
Disease: **ovarian cancer**; a female reproductive organ cancer that is located in the ovary. |
| Compound *causes* Side Effect | Compound: **cefaclor**; semisynthetic, broad-spectrum antibiotic derivative of cephalexin.
Side Effect: **tubulointerstitial nephritis**; *no description* | Compound: **perflutren**; a diagnostic medication to improve contrast in echocardiograms.
Side Effect: **palpitations**; irregular and/or forceful beating of the heart. |

Table 3: Examples from Hetionet where one model ranks the shown positive pair considerably higher than the other. LMs often perform better when there is semantic relatedness between head and tail text, but can be outperformed by a KGE model when head/tail entity text is missing or unrelated. Entity descriptions cut to fit.

may be too technical or generic to be informative (*methylprednisolone*, *allergic rhinitis*), or where no description is available (*cefaclor*, *tubulointerstitial nephritis*).[7] Furthermore, Fig. 2 shows that KG-PubMedBERT outperforms the best KGE model on a substantial fraction of the test set examples for each dataset.[8] These observations motivate an approach that leverages the strengths of both types of models by identifying examples where each model might do better, which leads to our results for model integration.

## 4.2 Model Averaging and Routing

**Integrating pairs of models.** Table 2 shows that combining each class of models boosts results by a large relative improvement of 13–36% in MRR across datasets. Moreover, the best-performing combination always includes a KGE model and KG-PubMedBERT rather than two KGE models (Fig. 3), showing the unique benefit of using LMs to augment models relying on KG structure alone.

**Averaging vs. routing.** We also compare the router and input-dependent weighted average approaches of integrating a pair of models in Table 2, with the router-based approach outperforming the weighted average for the best KGE + LM pair on RepoDB and MSI. This presents routing as a promising alternative for integrating KGE and LM models. Since the gradient boosted decision trees (GBDT) router achieves the best validation set

---

7. Table 7 in the appendix shows the drop in performance when one or both entities are missing descriptions.
8. See MRR breakdown by relation type in Fig. 4 in the appendix.

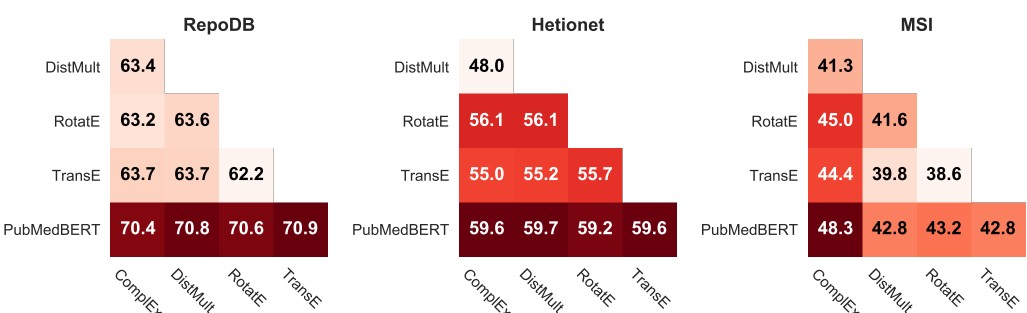

Figure 3: Test set MRR for all pairs of KG completion models using an MLP router. The best combination of a KGE model and KG-PubMedBERT always performs better than the best pair of KGE models, and for RepoDB and Hetionet all pairs involving KG-PubMedBERT outperform all KGE-only pairs.

performance in most cases across classifiers and integration methods, we use this method for combinations of more than two models, such as multiple LMs with a single KGE model.

**Integrating additional models.** The bottom of Table 2 shows results for three-model combinations. Adding a third model improves performance compared to the best pair of models, though whether the best third model is an LM or KGE model varies across datasets and metrics. Although there are diminishing returns to including a third model, the three-model combinations provide the best performance for RepoDB and MSI.

**Interpreting model routing.** We compute average feature gain for all datasets, using a GBDT router implemented with XGBoost [Chen and Guestrin, 2016] (see Fig. 5 in the appendix). We find that the most salient features are the ranking scores output by each model, which is intuitive as these scores reflect each model's confidence. Graph features like node degree and PageRank also factor into the classifier's predictions, as well as textual features such as entity text length and edit distance between entity names. General concepts such as *Hypertensive disease* and *Infection of skin and/or subcutaneous tissue* are central nodes for which we observe KGE models to often do better. KGE models also tend to do better on entities with short, non-descriptive names (e.g., *P2RY14*), especially when no descriptions are available. Generally, these patterns are not clear-cut, and non-linear or interaction effects likely exist. It remains an interesting challenge to gain deeper understanding into the strengths and weaknesses of LM-based and graph-based models.

### 4.3 Inductive KG Completion

For our inductive KG completion experiments, we use ComplEx as the KGE model and KG-PubMedBERT as our LM-based model, and compare the performance of each method to ComplEx with entity embeddings imputed using the method described in Section 2.4. We use either the untrained PubMedBERT or the fine-tuned KG-PubMedBERT as the LM for retrieving nearest-neighbor (NN) entities (see examples in Table 9 in the appendix). We also compare to DKRL [Xie et al., 2016], which constructs entity representations from text using a CNN encoder and uses the TransE scoring function. We use PubMedBERT's

|  | RepoDB | | | Hetionet | | | MSI | | |
|---|---|---|---|---|---|---|---|---|---|
|  | MRR | H@3 | H@10 | MRR | H@3 | H@10 | MRR | H@3 | H@10 |
| DKRL | 15.6 | 15.9 | 28.2 | 17.8 | 18.5 | 31.9 | 13.3 | 14.1 | 22.4 |
| KG-PubMedBERT | **38.8** | **43.4** | **67.5** | **21.6** | **22.3** | **42.8** | **20.2** | **21.7** | **32.2** |
| ComplEx | 0.8 | 0.4 | 1.6 | 3.6 | 0.7 | 2.8 | 0.5 | 0.1 | 0.4 |
| NN-ComplEx, frozen LM | 20.1 | 22.3 | 31.2 | 18.1 | 18.4 | 32.8 | 15.8 | 16.9 | 23.4 |
| NN-ComplEx, fine-tuned | 26.9 | 30.3 | 39.4 | 13.9 | 12.9 | 25.5 | 14.6 | 15.4 | 21.4 |

Table 4: Inductive KG completion results. NN-ComplEx refers to the version of ComplEx with unseen entity embeddings replaced using an LM to find the 1-nearest neighbor, either with PubMedBERT frozen or fine-tuned for KG completion (KG-PubMedBERT).

token embeddings as input to DKRL and train with the same multi-task loss. While other methods for inductive KG completion exist, such as those based on graph neural networks [Schlichtkrull et al., 2018, Vashishth et al., 2020, Bhowmik and de Melo, 2020], they require the unseen entity to have *known connections* to entities that were seen during training in order to propagate information needed to construct the new embedding. In our inductive experiments, we consider the more challenging setup where every test set triple has at least one entity with no known connections to entities seen during training, such that graph neural network-based methods cannot be applied. This models the phenomenon of rapidly emerging concepts in the biomedical domain, where a novel drug or protein may be newly studied and discussed in the scientific literature without having been integrated into existing knowledge bases.

As seen in Table 4, ComplEx unsurprisingly performs poorly as it attempts link prediction with random embeddings for unseen entities. DKRL does substantially better, with KG-PubMedBERT further increasing MRR with a relative improvement of 21% (Hetionet) to over 2x (RepoDB). Our strategy for replacing ComplEx embeddings for unseen entities performs comparably to or better than DKRL in most cases, with untrained PubMedBERT encodings generally superior to using KG-PubMedBERT's encodings. In either case, this simple strategy for replacing the untrained entity embeddings of a KGE model shows the ability of an LM to augment a structure-based method for KG completion that is typically only used in the transductive setting, even without using the LM to compute ranking scores.

## 5. Conclusion and Discussion

We perform the first empirical study of scientific language models (LMs) applied to biomedical knowledge graph (KG) completion. We evaluate *domain-specific* biomedical LMs, fine-tuning them to predict missing links in KGs that we construct by enriching biomedical datasets with textual entity descriptions. We find that LMs and more standard KG embedding models have complementary strengths, and propose a routing approach that integrates the two by assigning each input example to either type of model to boost performance. Finally, we demonstrate the utility of LMs in the inductive setting with entities not seen during training, an important scenario in the scientific domain with many emerging concepts.

Our work raises several directions for further study. For instance, several structural differences exist between general-domain and biomedical text that would be interesting to

explore in more depth and leverage more explicitly to improve KG completion performance. For example, entities with uninformative technical names – such as protein names that are combinations of numbers and letters (e.g., P2RY14) – appear very often in scientific KGs, and are likely related to the benefit of adding descriptions (Table 7, appendix). The surface forms of entity mentions in the biomedical literature on which the LMs were pretrained tend to be diverse with many aliases, while entities such as cities or people in the general domain often show less variety in their surface forms used in practice. This could potentially be challenging when trying to tap into the latent knowledge LMs hold on specific entities as part of the KG completion task, and likely requires LMs to disambiguate these surface forms to perform the task well. General-domain LMs are also trained on corpora such as Wikipedia which has "centralized" pages with comprehensive information about entities, while in the scientific literature information on entities such as drugs or genes is scattered across the many papers that form the training corpora for the LMs.

Previous work [Wang et al., 2021] has also observed that combining graph- and LM-based models improves KG completion results. We provide further analyses into this phenomenon based on textual and graph properties, but a deeper understanding of the strengths and weaknesses of each modality is needed. Interpreting neural models is generally a challenging problem; further work in our setting could help reveal the latent scientific knowledge embedded in language models. Importantly, our results point to the potential for designing new models that capitalize on both graph and text modalities, perhaps by injecting structured knowledge into LMs [Peters et al., 2019] or with entity-centric pretraining [Zemlyanskiy et al., 2021]. Finally, our findings provide a promising direction for biomedical knowledge completion tasks, and for literature-based scientific discovery [Swanson, 1986, Gopalakrishnan et al., 2019].

## Acknowledgments

This project is supported in part by NSF Grant OIA-2033558 and by the Office of Naval Research under MURI grant N00014-18-1-2670.

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

| Dataset | Link | Sources |
|---------|------|---------|
| RepoDB | http://apps.chiragjpgroup.org/repoDB/ | DrugBank UMLS |
| Hetionet | https://github.com/hetio/hetionet | DrugBank Disease Ontology Entrez SIDER MeSH |
| MSI | https://github.com/snap-stanford/multiscale-interactome | DrugBank Gene Ontology Entrez UMLS |

Table 5: Links and sources of entity names and descriptions for each dataset.

## Appendix A. Dataset Construction

### A.1 Sources

**RepoDB**   Drugs in RepoDB have statuses including *approved*, *terminated*, *withdrawn*, and *suspended*. We restrict our KG to pairs in the *approved* category.

**Hetionet**   was constructed using data from various publicly-available scientific repositories. Following Alshahrani et al. [2021], we restrict the KG to the *treats*, *presents*, *associates*, and *causes* relation types. This includes interactions between drugs and the diseases they treat, diseases and their symptoms, diseases and associated genes, and drugs and their side effects. We use this subset of the full Hetionet dataset to avoid scalability issues that arise when training large Transformer-based language models, inspired by benchmark datasets such as FB15K [Bordes et al., 2013], a subset of the Freebase knowledge base.

**MSI**   includes diseases and the proteins they perturb, drug targets, and biological functions designed to discover drug-disease treatment pairs through the pathways that connect them via genes, proteins, and their functions. We include all entities and relation types in the dataset.

We collect each of the datasets from the links listed in Table 5. For missing entity names and all descriptions, we write scripts to scrape the information from the resources listed above using the entity identifiers provided by each of the datasets.

### A.2 Transductive Splits

To construct transductive splits for each dataset, we begin with the complete graph, and repeat the following steps:

1. Randomly sample an edge from the graph.

2. If the degree of both nodes incident to the edge is greater than one, remove the edge.

3. Otherwise, replace the edge and continue.

The above steps are repeated until validation and test graphs have been constructed of the desired size while ensuring that no entities are removed from the training graph. We construct 80%/10%/10% training/validation/test splits of all datasets.

## A.3 Inductive Splits

To construct inductive splits for each dataset, we follow the procedure outlined in the "Technical Details" section of the appendix of Daza et al. [2021]. We similarly construct a 80%/10%/10% training/validation/test split of each dataset in the inductive setting.

## A.4 Negative Validation/Test Triples

In order to perform a ranking-based evaluation for each dataset in both the transductive and inductive settings, we generate a set of negative triples to be ranked against each positive triple. To generate negative entities to replace both the head and tail entity of each validation and test positive, we follow the procedure below:

1. Begin with the set of all entities in the knowledge graph.

2. Remove all entities that do not have the same entity type as the entity to be ranked against in the positive triple.

3. Remove all entities that would result in a valid positive triple in either the training, validation, or test sets.

4. Randomly sample a fixed set of size $m$ from the remaining set of entities.

We use a value of $m = 500$ for RepoDB and MSI, and a value of $m = 80$ for Hetionet (due to the constraints above, the minimum number of valid entities remaining across positive triples for Hetionet was 80). Using a fixed set of entities allows for fair comparison when assessing performance of subsets of the test set, such as when examining the effect of subsets where descriptions are present for neither, one, or both entities (Table 7).

## Appendix B. Training

### B.1 Transductive Setting

For all individual models, we train the models on the training set of each dataset while periodically evaluating on the validation set. We save the model with the best validation set MRR, then use that model to evaluate on the test set. We also perform hyperparameter tuning for all models, and use validation set MRR to select the final set of hyperparameters for each model.

#### B.1.1 KNOWLEDGE GRAPH EMBEDDINGS

We use the max-margin ranking loss for all KGE methods. We use a batch size of 512 for all models. We train models for 10,000 steps (958 epochs) on RepoDB, 50,000 steps (205 epochs) on Hetionet, and 50,000 steps (66 epochs) on MSI. We evaluate on the validation set every 500 steps for RepoDB and 5,000 steps for Hetionet and MSI. We use the Adam optimizer for training. We perform a hyperparameter search over the following values:

- Embedding dimension: 500, 1000, 2000

- Margin for max-margin loss: 0.1, 1

- Learning rate: 1e-3, 1e-4

- Number of negative samples per positive: 128, 256

- Parameter for L3 regularization of embeddings: 1e-5, 1e-6

### B.1.2 LANGUAGE MODELS

**Pretrained scientific LMs.** We explore various pretrained LMs, with their initialization, vocabulary, and pretraining corpora described below. In particular, we study a range of LMs trained on different scientific and biomedical literature, and also on clinical notes.

- **BioBERT** [Lee et al., 2020] Initialized from BERT and using the same general domain vocabulary, with additional pretraining on the PubMed repository of scientific abstracts and full-text articles.

- **Bio+ClinicalBERT** [Alsentzer et al., 2019] Initialized from BioBERT with additional pretraining on the MIMIC-III corpus of clinical notes.

- **SciBERT** [Beltagy et al., 2019] Pretrained from scratch with a domain-specific vocabulary on a sample of the Semantic Scholar corpus, of which biomedical papers are a significant fraction but also papers from other scientific domains.

- **PubMedBERT** [Gu et al., 2020] Pretrained from scratch with a domain-specific vocabulary on PubMed. We apply two versions of PubMedBERT, one trained on PubMed abstracts alone (PubMedBERT-abstract) and the other on abstracts as well as full-text articles (PubMedBERT-fulltext).

We also use **RoBERTa** [Liu et al., 2019] – pretrained from scratch on the BookCorpus, English Wikipedia, CC-News, OpenWebText, and Stories datasets – as a strongly-performing general domain model for comparison. For all LMs, we follow Kim et al. [2020] and use the multi-task loss consisting of binary triple classification, multi-class relation classification, and max-margin ranking loss, with a margin of 1 for the max-margin loss. For triple classification, given the correct label $y \in \{0, 1\}$ (positive or negative triple) we apply a linear layer to the [CLS] token representation $v$ to output the probability $p$ of the triple being correct as $p = \sigma(W_{\text{triple}}v)$, and use the binary cross entropy loss $\mathcal{L}_{\text{triple}}(x) = -y \log(p) - (1 - y) \log(1 - p)$. For relation classification over $R$ relation types, we apply a linear layer to $v$ to calculate a probability distribution $q$ over relation classes with $q = \text{softmax}(W_{\text{rel}}v)$, and use the cross entropy loss with one-hot vector $y \in \{0, 1\}^R$ as the correct relation label: $\mathcal{L}_{\text{rel}}(x) = -\sum_{i=1}^{R} y_i \log q_i$. The final loss is the equally-weighted sum of all three losses: $\mathcal{L}(x) = \mathcal{L}_{\text{rank}}(x) + \mathcal{L}_{\text{triple}}(x) + \mathcal{L}_{\text{rel}}(x)$.

We train for 40 epochs on RepoDB, and 10 epochs on Hetionet and MSI. We evaluate on the validation set every epoch for RepoDB, and three times per epoch for Hetionet and MSI. For RepoDB, Hetionet, and MSI we use 32, 16, and 8 negative samples per positive, respectively. We use the Adam optimizer for training. We perform a hyperparameter search over the following values:

- Batch size: 16, 32

- Learning rate: 1e-5, 3e-5, 5e-5

### B.1.3 INTEGRATED MODELS

**Global weighted average.** For the global weighted average, we compute ranking scores for positive and negative examples as the weighted average of ranking scores output by all KG completion models being integrated. Specifically, for a set of ranking scores $s_1, \ldots, s_k$ output by $k$ models for an example, we learn a set of weights $\boldsymbol{\alpha} = [\alpha_1, \ldots, \alpha_k]$ to compute the final ranking score as $s = \sum_{i=1}^{k} \alpha_i s_i$, where the same weight vector $\boldsymbol{\alpha}$ is used for all examples. We search for each $\alpha_i$ over the grid [0.05, 0.95] with steps of 0.05, ensuring that all $\alpha_i$'s sum to 1. We choose values that maximize validation set MRR, then apply them to the test set.

**Router.** For the router-based method, we train a classifier to select a single model out of a set of KG completion models to use for computing ranking scores for a positive example and its associated negatives. The class to be predicted for a particular example corresponds to which model performs best on that example (i.e., gives the best rank), with an additional class for examples where all models perform the same. We explore a number of different classifiers, including logistic regression, decision tree, gradient boosted decision tree (GBDT), and multilayer perceptron (MLP), finding that GBDT and MLP classifiers perform the best. As input to the classifier, we use a diverse set of features computed from each positive example (listed in Table 6) as well as each model's ranking score for the positive example. Classifiers are trained on the validation set and evaluated on the test set for each dataset. We additionally perform hyperparameter tuning over the following values for each classifier:

Logistic regression:

- Penalty: L1, L2

- Regularization parameter: 9 values evenly log-spaced between 1e-5 and 1e3

Decision tree:

- Max depth: 2, 4, 8

- Learning rate: 1e-1, 1e-2, 1e-3

GBDT:

- Number of boosting rounds: 100, 500, 1000

- Max depth: 2, 4, 8

- Learning rate: 1e-1, 1e-2, 1e-3

MLP:

- Number of hidden layers: 1, 2

- Hidden layer size: 128, 256

- Batch size: 64, 128, 256

- Learning rate: 1e-1, 1e-2, 1e-3

We perform five-fold cross-validation on the validation set and use validation set accuracy to choose the best set of hyperparameters for each classifier. We use Scikit-Learn [Pedregosa et al., 2011] to implement the logistic regression and MLP classifiers, and XGBoost [Chen and Guestrin, 2016] to implement the decision tree and GBDT classifiers, using default parameters other than the ones listed above.

**Input-dependent weighted average.** The input-dependent weighted average method of integrating KG completion models operates similarly to the global weighted average, except that the set of weights can vary for each positive example and are a function of its feature vector (the same set of weights is used for all negative examples used to rank against each positive example). We train an MLP to output a set of weights that are then used to compute a weighted average of ranking scores for a set of KG completion models. The MLP is trained on the validation set and evaluated on the test set for each dataset. We use the max-margin ranking loss with a margin of 1. In order to compare to the MLP trained as a router, we train the MLP using the Adam optimizer [Kingma and Ba, 2015] for 200 epochs with early stopping on the training loss and a patience of 10 epochs (the default settings for an MLP classifier in Scikit-Learn). We perform a hyperparameter search over the following values (matching the values for the MLP router where applicable):

- Number of hidden layers: 1, 2

- Hidden layer size: 128, 256

- Batch size: 64, 128, 256

- Learning rate: 1e-1, 1e-2, 1e-4

- Number of negatives (for max-margin loss): 16, 32

We select the best hyperparameters by MRR on a held-out portion of the validation set.

**Features for integrated models.** Both the router and input-dependent weighted average methods of model integration use a function to outputs weights based on a feature vector of an example. A complete list of the features used by each method can be found in Table 6. We also use the ranking score for the positive example from each KG completion model being integrated as additional features.

## B.2 Inductive Setting

### B.2.1 Knowledge Graph Embeddings and Language Models

For the KGE and LM models, we follow the same training procedure for the inductive splits as for the transductive splits. We perform hyperparameter tuning over the same grids of hyperparameters, periodically evaluate on the validation set and save the checkpoint with the best validation set MRR, and use the set of hyperparameters corresponding to the highest validation set MRR to evaluate on the test set.

| | |
|---|---|
| entity type | length of text in chars. |
| relation type | presence of word "unknown" in name/desc. |
| head/tail node in-/out-degree | missing desc. |
| head/tail node PageRank | number/ratio of punctuation/numeric chars. |
| Adamic-Adar index of edge | tokens-to-words ratio of entity name/desc. |
| edit dist. between head/tail entity names | |

Table 6: Complete list of features used by router classifiers.

### B.2.2 DKRL

In addition to the KGE and LM-based methods, we also train DKRL [Xie et al., 2016] for inductive KG completion as another text-based baseline for comparison. DKRL uses a two-layer CNN encoder applied to the word or subword embeddings of an entity's textual description to construct a fixed-length entity embedding. To score a triple, DKRL combines its entity embeddings constructed from text with a separately-learned relation embedding using the TransE [Bordes et al., 2013] scoring function. The original DKRL model uses a joint scoring function with structure-based and description-based components; we restrict to the description-based component as we are applying DKRL in the inductive setting. We use PubMedBERT subword embeddings at the input layer of the CNN encoder, encode entity names and descriptions, and apply the same multi-task loss as for the LM-based models. To apply the triple classification and relation classification losses, for head and tail entity embeddings $\mathbf{h}$ and $\mathbf{t}$, we apply a separate linear layer for each loss to the concatenated vector $[\mathbf{h}; \mathbf{t}; |\mathbf{h} - \mathbf{t}|]$, following previous work on models that use a bi-encoder to construct entity or sentence representations [Wang et al., 2021, Reimers and Gurevych, 2019]. We use the same number of training epochs and number of negatives per positive for DKRL as for the LM-based methods on each dataset. We use a batch size of 64, and perform a hyperparameter search over the following values:

- Learning rate: 1e-3, 1e-4, 1e-5

- Embedding dimension: 500, 1000, 2000

- Parameter for L2 regularization of embeddings: 0, 1e-3, 1e-2

## Appendix C. Additional Results

### C.1 Transductive Setting, Individual Models

**Missing entity descriptions.** Table 7 shows test set MRR for KG-PubMedBERT on each dataset broken down by triples with either both, one, or neither entities having available descriptions. Across datasets, performance clearly degrades when fewer descriptions are available to provide context for the LM to generate a ranking score.

**Relation-level performance.** Figure 4 shows test set MRR broken down by relation for the datasets with multiple relation types (Hetionet and MSI). KG-PubMedBERT performs better on all relation types except compound-side effect for Hetionet, and on the function-function relation for MSI.

| #entities with desc. in pair | MRR | | |
|---|---|---|---|
| | **RepoDB** | **Hetionet** | **MSI** |
| None | N/A | 25.6 | 25.1 |
| One | 59.5 | 43.6 | 25.4 |
| Both | 63.7 | 52.6 | 37.3 |

Table 7: Effect of descriptions on KG-PubMedBERT test set MRR.

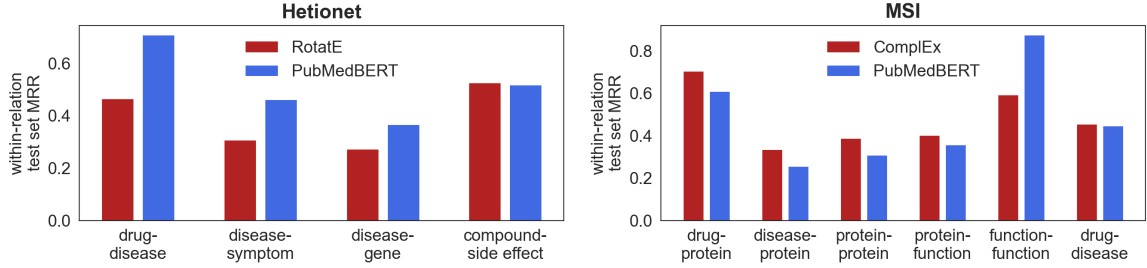

Figure 4: Test set MRR for the best KGE model compared to KG-PubMedBERT broken down by relation type for Hetionet and MSI.

## C.2 Transductive Setting, Integrated Models

| | **RepoDB** | **Hetionet** | **MSI** |
|---|---|---|---|
| Global avg. | 70.4 | 55.8 | 42.1 |
| Input-dep. avg. | 65.9 | **70.3** | 40.6 |
| Router | **70.6** | 59.7 | **48.5** |

Table 8: Test set MRR for the best pair of a KGE model and KG-PubMedBERT for different methods of model integration.

## C.3 Inductive Setting

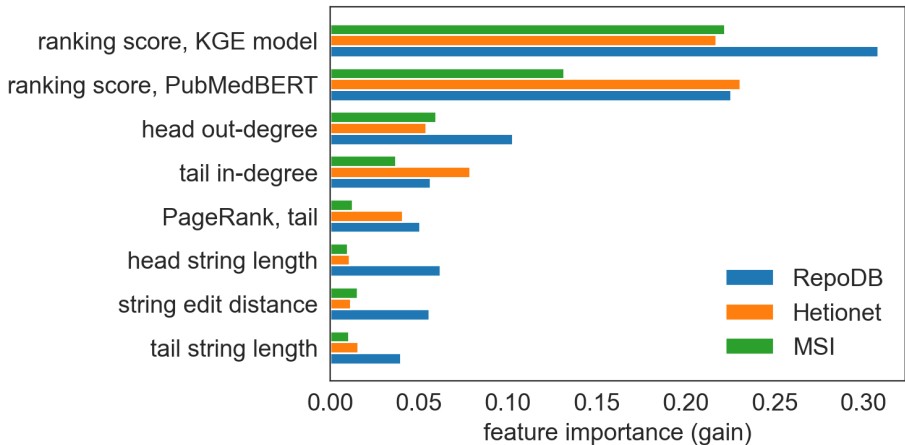

Figure 5: Feature importances for GBDT router for a selection of most important features. Ranking scores output by each model tend to be the most important, with other graph- and text-based features also contributing.

| Imputation Model with Better Ranking | Unseen Entity | KG-PubMedBERT nearest neighbors | PubMedBERT nearest neighbors |
|---|---|---|---|
| PubMedBERT | eye redness | skin burning sensation, skin discomfort | conjunctivitis, throat sore |
| | ecchymosis | gas, thrombophlebitis | petechiae, macule |
| | estrone | vitamin a, methyltestosterone | estriol, calcitriol |
| KG-PubMedBERT | keratoconjunctivitis | conjunctivitis allergic, otitis externa | enteritis, parotitis |
| | malnutrition | dehydration, anaemia | meningism, wasting generalized |
| | congestive cardiomyopathy | diastolic dysfunction, cardiomyopathy | carcinoma breast, hypertrophic cardiomyopathy |

Table 9: Samples of unseen entities and their nearest neighbors found by KG-PubMedBERT and PubMedBERT, for test set examples in the Hetionet inductive split where the Pub-MedBERT neighbor performs better than the KG-PubMedBERT neighbor (first three) and vice versa (last three). Each LM offers a larger improvement per example when its nearest neighbor is more semantically related to the unseen entity.