# OpenReview forum: "Scientific Language Models for Biomedical Knowledge Base Completion: An Empirical Study"
_AKBC.ws/2021/Conference — AKBC 2021_

### Official Review · Reviewer_fvNB · 2021-07-20
**A comprehensive emprical study**

**Rating:** 8
**Confidence:** 3

**Review:**

**Summary**:

This paper presented a comprehensive empirical study of applying various scientific language models to the biomedical knowledge graph completion task. The authors found that language models and knowledge graph embedding models have complementary strengths, and further proposed a routing method, which is able to integrate these two models. Domain-specific biomedical language models are carefully evaluated.

**Reasons to accept**:

1. The paper is well-organized and nicely presented. A wide range of methods including the very recent ones are surveyed. I am not an expert in this research direction, and I learned a lot from this paper.

2. Extensive experiments are conducted to investigate the existing models under different settings and verify the effectiveness of the proposed routing method under the transductive settings. Details of the experimental settings such as datasets, hyper-parameters, baselines are included.

**Reasons to reject**:

No strong reasons for rejection. However, as mentioned by the other reviewer, graph neural network-based models should also be considered for discussions and comparisons.

**Questions**: ​

1.Though investigating general-domain language models is out of the scope of the paper, I am wondering if the proposed method for domain-specific LM also works for general-domain language models?

2.For biomedical knowledge base completion, do we have any domain-specific issues when applying LM to it?

3.The last question is whether integrating LM-based models with KG embedding models is an under-explored research question for the general-domain settings.

---

> ### Author Response · Authors · 2021-07-29
> **Response to Reviewer fvNB**
>
> Thank you for your feedback. We invite you to consider our response to reviewer qESR regarding comparison to graph neural network-based models. We address your other questions below:
>
> 1. We focus on the biomedical domain for the purpose of this study as the use of LMs for KG completion has been underexplored in this important setting. However, we have demonstrated that some techniques – such as using a router rather than a weighted average for combining LMs and KGE models – can sometimes work better than approaches that have been used in the general domain. This certainly opens up the possibility that these or other extensions to existing techniques may improve performance on general domain KG completion benchmarks as well, which we leave as a direction for future studies.
>
>
> 2. One main difference between biomedical and general domain KGs is the domain-specific terminology and jargon that exists in names and descriptions of biomedical entities. Much like in general natural language understanding tasks, we demonstrate that LMs pretrained on domain-specific texts are crucial for biomedical KG completion, with the biomedical LMs we explore outperforming a general domain LM (RoBERTa) in nearly all cases. In addition, we believe there are more structural differences that would be interesting to explore in more depth. For example, entities with uninformative technical names – such as protein names that are combinations of numbers and letters (e.g., P2RY14) – appear very often in scientific KGs, and are likely related to the benefit of adding descriptions (Table 6, appendix). The surface forms of entity mentions in the biomedical literature (on which the LMs were pre-trained) tend to be diverse with many aliases (e.g., here is a [list of aliases](https://www.genecards.org/cgi-bin/carddisp.pl?gene=ERBB2) for one gene), while entities such as cities (e.g., “Paris”) or people (e.g., “Albert Einstein”) in the general domain often show less variety in their surface forms used in practice. This could potentially be challenging when trying to tap into the latent knowledge LMs hold on specific entities as part of the KG completion task, and likely requires LMs to disambiguate these surface forms to perform the task well. General-domain LMs are also trained on corpora such as Wikipedia which has “centralized” pages with comprehensive information about entities, while in the scientific literature information on entities such as drugs or genes is scattered across the many papers that form the training corpora for the LMs.
>
>
> 3. We definitely think this is an underexplored area where much more work can be done. There has been some work on general domain benchmarks that computes a weighted average of the ranking scores from each type of model [1-3]. We show in this work that adaptively routing each example to a single model can sometimes be preferable to using an average over all models. We also demonstrate that integrating more than two LM/KGE models can yield additional benefits, something which has not been explored previously and which emphasizes the complementary strengths of even different domain-specific LMs applied to the same dataset. Finally, we briefly explore which features are related to whether an LM or a KGE model will perform better on a particular example, although our findings are not always clear-cut and characterizing this further is an open question. Through these contributions, we extend the understanding of how LM and KGE models can be effectively integrated to improve KG completion.
>
>
> [1] Toutanova, K., Chen, D., Pantel, P., Poon, H., Choudhury, P., & Gamon, M. (2015). Representing Text for Joint Embedding of Text and Knowledge Bases. EMNLP.
>
> [2] Xie, R., Liu, Z., Jia, J., Luan, H., & Sun, M. (2016). Representation Learning of Knowledge Graphs with Entity Descriptions. AAAI.
>
> [3] Wang, B., Shen, T., Long, G., Zhou, T., & Chang, Y. (2021). Structure-Augmented Text Representation Learning for Efficient Knowledge Graph Completion. WWW.

---

> > ### Comment · Reviewer_fvNB · 2021-07-29
> > **Thanks for the reply**
> >
> > Thanks for the detailed reply. I would like to keep my score for the paper.

---

### Official Review · Reviewer_qESR · 2021-07-21
**Strong empirical study for transductive setting, but weaker in the inductive setting**

**Rating:** 7
**Confidence:** 5

**Review:**

This paper proposes a method for link prediction in biomedical knowledge graphs that combines the predictions of knowledge graph embedding models with pre-trained language models. The main contribution of this paper is a routing method that acts as a switch between a set of KG embedding models and language models (that are fine-tuned for the link prediction task) and outputs the best model’s prediction for a given triple (h, r, t).

There are similar methods to this paper proposed in the general KG completion task (cited in this paper). This paper extends those methods to biomedical knowledge graphs. The key difference is that existing methods use a weighted average of different model’s predictions whereas the proposed model chooses one of the model’s predictions using a routing method. Empirical results show that this approach is effective for transductive settings. The main finding of the paper is that language models and knowledge graph embedding models are complementary.

**Reasons to accept**
1. The paper is well-written and performs a thorough empirical study by considering a wide range of methods and various combinations of them.
2. Performed detailed analysis of the model’s predictive performance.

**Reasons to reject**
1. The proposed model performs much worse than the fine-tuned LM model in an inductive setting. Also, the reason to choose the nearest neighbor of the unseen entity is not well justified.  Since the embedding of the unseen entity is obtained using a fine-tuned LM, why can’t we directly use that?
2. There are other strong baselines for inductive settings such as R-GCN [1], CompGCN [2], and [3] that are not considered for comparisons. Also, integrating these models into the proposed method might further improve the performance.

**Comments**
Please mention the number of relations in each dataset in Table 1.

**Missing references**

[1] Schlichtkrull, M.S., Kipf, T.N., Bloem, P., van den Berg, R., Titov, I., Welling, M.: Modeling relational data with graph convolutional networks. In: Proceedings of ESWC 2018. LNCS, vol. 10843, pp. 593–607. Springer (2018)

[2] Vashishth, S., Sanyal, S., Nitin, V., Talukdar, P.: Composition-based multi-relational graph convolutional networks. In: International Conference on Learning Representations (2020)

[3] R Bhowmik, G de Melo. Explainable Link Prediction for Emerging Entities in Knowledge Graphs.  International Semantic Web Conference, 2020

---

> ### Author Response · Authors · 2021-07-29
> **Response to Reviewer qESR**
>
> Thank you for your constructive and detailed feedback. We address your concerns below:
>
> 1. The purpose of the experiment in the inductive setting was not to propose a new model, but to study a simple KGE baseline to compare with the fine-tuned LM. Since standard KGE models can only be applied in the transductive setting, this baseline uses LM representations to find the most similar entity in the training set based on textual information, and uses that entity’s structural embedding learned by the KGE model for computing the ranking score. Since the entity embedding computed by the LM is not a KG embedding that has been trained with the ComplEx scoring function, it cannot be directly plugged in to replace the missing ComplEx embedding, although it can still be used to retrieve the embedding of the most semantically-related “nearest neighbor” entity. We don’t expect this simple baseline to perform better than the fine-tuned LM, and the fine-tuned LM definitely can directly be used in this setting. However, it is interesting that despite its simplicity the nearest-neighbor baseline can actually outperform a more sophisticated baseline model (DKRL) that relies solely on textual information. This is an indication that combining structural and textual information can provide benefits in the inductive setting as well, and can be explored further in the future.
>
>
> 2. While graph neural network-based methods like the ones you’ve proposed can certainly be used in the inductive setting to construct an embedding for an unseen entity, they require the unseen entity to have _known connections_ to entities that were seen during training in order to propagate information needed to construct the new embedding. In our inductive experiments, we consider the more challenging setup where every test set triple has at least one entity with no known connections to entities seen during training, such that graph neural network-based methods cannot be applied. This is meant to model the phenomenon of rapidly emerging concepts in the biomedical domain, where a novel drug or protein may be newly studied and discussed in the scientific literature without having been integrated into existing knowledge bases. We will clarify this point in the section on inductive KG completion and cite the methods you’ve mentioned.
>
> As to your comment, we will be sure to include the number of relations for each dataset in Table 1. We hope our responses addressed your concerns, and we welcome any additional clarifying questions.

---

### Official Review · Reviewer_pTkv · 2021-07-22
**nice contribution**

**Rating:** 8
**Confidence:** 4

**Review:**

The paper uses domain-specific pretrained language models to biomedical knowledge graph completion.

- I like the research problem of the paper. Pretrained language models are definitely a way to extract knowledge from unstructured texts. It would be great if we could find a way to extract the knowledge from LM to structured or semi-structured stuff. Using LMs for knowledge graph completion is a clever way, which is easy to evaluate.

- Why is the domain-specific task evaluated in the paper? How about the open-domain setup?

- It's good to see the proposed method is complementary to previous methods. There would be some opportunities to better integrate these methods together.

- In terms of interpretation, there is a related paper "Knowledge Neurons in Pretrained Transformers", which also tries to align the neurons in LMs and knowledge.

- Is this a good way to evaluate pretrained LMs? A metric could be defined for this.

---

> ### Author Response · Authors · 2021-07-29
> **Response to Reviewer pTkv**
>
> Thank you for your detailed feedback. We evaluate the KG completion task in the biomedical domain specifically as applying LMs to biomedical KG completion is still underexplored despite the importance of this domain, and our empirical analysis demonstrates a variety of ways in which LMs can be effectively applied in this domain (although some of these techniques may also transfer to the general domain – see our response to reviewer fvNB).
>
> The “Knowledge Neurons” approach could definitely be combined with the datasets we’ve explored in this work, to examine more specifically which network parameters encode what biomedical knowledge and modify that knowledge if desired.
>
> With regard to evaluating pretrained LMs for factual knowledge, one dominant paradigm for this kind of evaluation is probing the LM using a cloze task such as with the LAMA dataset [1]. The KG completion task could similarly be used to evaluate what factual knowledge is stored in an LM’s parameters, and its ability to identify stored knowledge could be compared to the results of a more LAMA-style evaluation in future work.
>
>
> [1] Petroni, F., Rocktäschel, T., Lewis, P., Bakhtin, A., Wu, Y., Miller, A.H., & Riedel, S. (2019). Language Models as Knowledge Bases? EMNLP.

---

### Decision · Program_Chairs · 2021-08-17

**Decision:**

Accept

**Comment:**

The paper presents an in-depth evaluation of biomedical language models to the task of biomedical knowledge graph completion. A main takeaway is that language models and knowledge graph embedding models capture complementary signals. The paper then proposes a routing method to switch between the two, obtaining good results. The paper presents a thorough empirical study by considering a wide range of methods.